# Nordic study on human milk fortification in extremely preterm infants: a randomised controlled trial — the N-forte trial

Georg Bach Jensen [1,2] Fredrik Ahlsson [3] Magnus Domellöf [4]
Anders Elfvin [5,6] Lars Naver [7,8] Thomas Abrahamsson [1,2]

For numbered affiliations see end of article.

**Correspondence to**
Dr Thomas Abrahamsson;
thomas.abrahamsson@liu.se

## ABSTRACT

**Introduction** The mortality rate of extremely low gestational age (ELGA) (born <gestational week 28+0) infants remains high, and severe infections and necrotising enterocolitis (NEC) are common causes of death. Preterm infants receiving human milk have lower incidence of sepsis and NEC than those fed a bovine milk-based preterm formula. Despite this, fully human milk fed ELGA infants most often have a significant intake of cow's milk protein from bovine-based protein fortifier. The aim of this study is to evaluate whether the supplementation of human milk-based, as compared with bovine-based, nutrient fortifier reduces the prevalence of NEC, sepsis and mortality in ELGA infants exclusively fed with human milk.

**Methods and analysis** A randomised-controlled multicentre trial comparing the effect of a human breast milk-based fortifier with a standard bovine protein-based fortifier in 222–322 ELGA infants fed human breast milk (mother's own milk and/or donor milk). The infants will be randomised to either fortifier before reaching 100 mL/ kg/day in oral feeds. The intervention, stratified by centre, will continue until the target postmenstrual week 34+0. The primary outcome is a composite of NEC, sepsis or death. Infants are characterised with comprehensive clinical and nutritional data collected prospectively from birth until hospital discharge. Stool, urine, blood and breast milk samples are collected for analyses in order to study underlying mechanisms. A follow-up focusing on neurological development and growth will be performed at 2 and 5.5 years of age. Health economic analyses will be made.

**Ethics and dissemination** The study is conducted according to ICH/GCP guidelines and is approved by the regional ethical review board in Linköping Sweden (Dnr 2018/193-31, Dnr 2018/384-32). Results will be presented at scientific meetings and published in peer-reviewed publications.

**Trial registration number** The study was registered with ClinicalTrials.gov NCT03797157, 9 January 2019.

## INTRODUCTION
### Extremely preterm infants are at risk of severe morbidity and mortality

Although neonatal care has improved markedly during the last decades, mortality and

---

## Strengths and limitations of this study

▶ A randomised controlled multicentre trial on human milk-based fortification in extremely low gestational age infants powered to detect a reduction in sepsis, NEC and mortality as a composite outcome.

▶ Several assessments of outcomes will be made blinded, such as the radiological assessment of NEC.

▶ Both study groups will receive exclusively enteral feeding with breast milk and thereby not differ, besides the different breast milk fortification.

▶ A significant positive result of this trial would justify implementation of human milk-based fortifiers in the neonatal intensive care units.

▶ Targeted fortification of breast milk and the difference in nutrient content between the study product and the standard fortification made it impossible to blind the study after randomisation.

---

morbidity remain high with a 1-year mortality of 23% in extremely low gestational age (ELGA) neonates in affluent countries.[1] It is well known that nutrition is a key factor for survival, clinical course and later outcome of high-risk preterm infants; and of particular relevance to this trial is the evidence for beneficial roles of human milk in neonatal care, underpinning current recommendations for its use.[2–4] Extremely preterm infants (born before 28 weeks of completed gestation) are initially fed both enterally and parenterally while enteral feed tolerance develops. The primary choice is the infant's own mother's milk. Preterm formula or donor breast milk can be used when mothers provide insufficient amounts of breast milk, and in the Nordic countries, donor breast milk is routinely used for preterm infants up to 32–34 weeks postmenstrual age. A growing literature supports the use of human milk for feeding preterm infants, in particular for the smallest and most vulnerable preterm infants.

## Reduction of necrotising enterocolitis (NEC)

NEC remains a significant problem in neonates with an incidence between 4% and 15% in very low birth weight (<1500 g) infants.[1 5 6] The pathogenesis is multifactorial, and the disease leads to intestinal inflammation, ischaemia and necrosis. The main risk factors besides prematurity are formula feeding, hypoxic–ischaemic injury and abnormal bacterial gut colonisation.[7 8] NEC frequently results in death (around 20%–30%) or need for surgical resection of necrotic gut which may lead to short bowel syndrome. Further, NEC is a major adverse factor for neurological impairment.[9 10] The health economic cost of NEC is large.[11 12] The most recent Cochrane meta-analysis, including 8 randomised controlled trials with a total of 1605 preterm infants, shows that human milk, as compared with formula, reduces the risk of NEC by a factor 1.9 but it is not clear whether this effect is due to protective factors in human milk or the detrimental effects of cow's milk protein in formula.[13]

## Reduction in neonatal sepsis

Neonatal late onset sepsis remains a life-threatening and common problem in neonatal care. The incidence is 35%–40% in Swedish ELGA infants (www.snq.se). Growing evidence links sepsis to the mode of feeding. Even though no significant effect of human milk on sepsis was shown in the latest meta-analysis,[13] observational studies have demonstrated that human milk intake is associated with less sepsis.[14–17]

## Long-term impact of human milk in neonatal care

Breast-fed infants have generally higher intelligence quotient scores and lower risk of obesity at school age compared with formula fed.[18] Observational studies have suggested beneficial effects of human milk in preterm infants on health outcomes at school age, including reduced low-density/high-density lipoprotein cholesterol ratio, diastolic blood pressure, insulin resistance and likelihood of obesity/overweight in adolescence,[19–22] higher bone mass in adolescence,[23] higher cognitive performance in childhood[24 25] and increased brain size and white matter in adolescence.[26]

## Nutritional inadequacy of breast milk for preterms: implications

It is well recognised, that despite the health benefits of human milk, it does not meet the nutritional needs of ELGA infants whose rates of somatic and brain growth far exceed that of term neonates. Poor growth in ELGA infants is associated with worse subsequent neurological development.[27] In order to meet the nutritional needs of enterally fed preterm infants, protein containing fortifiers are therefore required, and these are generally cow's milk-based. While cow's milk-based fortifiers are routinely used in modern neonatal care to provide adequate nutrition, there is some evidence, indicating that cow's milk-based fortifiers may have negative effects when compared with human milk-based fortifiers.[28 29]

## Clinical studies on human milk-based breast milk nutrient fortifiers

Three randomised trials have examined the impact of human milk-based versus bovine-based diets[29–31] of which only one study was truly designed to evaluate the impact of a human milk-based fortifier.[31] One examined the impact of replacing cow's milk-based preterm formulas and fortifiers with donor milk and a human milk-based fortifier in 207 infants.[29] Infants fed the human milk-based diet had a lower NEC rate (5.4% vs 15.9%) and lower risk of NEC surgery (1.4% vs 11.6%). A post hoc subgroup analysis of infants not receiving preterm formula (n=114) showed a 4.2-fold increased risk of NEC and a 5.1-fold increased risk of NEC surgery or death in infants receiving a cow's milk-based, as opposed to a human milk-based, fortifier.[28] In a Canadian randomised controlled trial, on human milk-based vs cow's milk-based nutrient fortifier in 127 preterm infants with a birth weight <1250 g, the infants were exclusively fed with breast milk.[31] The primary outcome, feeding tolerance, was similar in the two study groups. Although the trial was not powered to demonstrate a significant effect of severe complications such as NEC and late-onset sepsis, there was a significant reduction of retinopathy of prematurity (ROP) (1.6% vs 10.2%, p=0.04) in infants receiving the human milk-based fortifier.

## Objectives

The primary objective is to evaluate if the supplementation of a human milk-based fortifier reduces the severe complications of NEC, sepsis and mortality as compared with bovine protein-based fortifier in extremely preterm infants fed only human breast milk (mother's own milk and/or donor milk). Secondary objectives are to evaluate if a human milk-based fortifier supplementation reduces other severe complications such as bronchopulmonary dysplasia (BPD), ROP and neurological impairment. Possible mechanisms underlying the effect of a human milk-based fortifier will be analysed in blood, stool, urine and breast milk samples. Health economic analyses will be made to evaluate the costs and benefits of an introduction of human milk-based fortifier in neonatal intensive care units (NICUs).

## METHODS

### Design of the N-forte study

This is a randomised controlled multicentre superiority trial comparing diet supplementation with human breast milk-based nutrient fortifier and standard bovine protein-based nutrient fortifier in extremely preterm infants exclusively fed with human breast milk (protocol V.2020/V.4, 25 March 2020).

### Setting and participants

Enrolment of extremely preterm infants (born before gestational week 28+0) will be made at 7 level III NICUs in Sweden by an attending physician. The 7 level III NICUs

**Box 1  Inclusion and exclusion criteria**

**Inclusion criteria**
► Gestational age at birth 22+0–27+6; based on prenatal ultrasonography
► Enteral feeds < 100 mL/kg/day at the day of randomisation
► Written informed consent from the legal guardians of the infant
► The home clinic of the infant has the logistics of maintaining the intervention until week 34+0

**Exclusion criteria**
► Lethal or complicated malformation known at the time of inclusion
► Chromosomal anomalies known at the time of inclusion
► No realistic hope for survival at the time of inclusion
► Gastrointestinal malformation known at the time of inclusion
► Abdominal surgery before the time of inclusion
► Participation in another intervention trial aiming at having an effect on growth, nutrition, feeding intolerance or severe complications such as NEC and sepsis
► Infants having nutrient fortifier or formula prior to randomisation

NEC, necrotising enterocolitis.

and 17 level I–II neonatal units in their catchment area regions agreed to participate insuring the continuation in the event of a transfer from one hospital to another during the study period.

## Inclusion

Inclusion and exclusion criteria are listed in box 1. A written informed consent will be obtained from the legal guardians before inclusion (online supplemental material). The allocation will be concealed before inclusion, but after randomisation the study is not blinded.

## Randomisation

Infants are randomised 1:1 to receive either a human breast milk-based fortifier or a standard bovine protein-based nutrient fortifier before oral feeds have reached 100 mL/kg/day. Randomisation is based on the following stratification variables: primary enrolment site, gestational week (22+0–24+6 or 25+0–27+6), singleton/twin and gender. Twins are randomised together and therefore receive the same nutritional protocol. An adaptive randomisation scheme is used based on the method of minimisation. This includes a biased-coin randomisation scheme as needed in the adaptive scheme.[32] A web-based randomisation service centre is used: Randomize.net (Interrand, Ottawa, Ontario, Canada).

## Intervention

The active group will receive a human milk-based nutrient fortifier (Humavant +6, Prolacta Bioscience, California, USA). The nutrient content is displayed in online supplemental table 1. The control group will receive the standard bovine milk-based nutrient fortifier of the particular NICU including the infant. Adhering to the Swedish national guidelines, the intervention starts before the enteral feeds have reached 100 mL/kg/day. Any milk fortifiers must not be prescribed prior

to inclusion. The attending physician and/or dietitian prescribes the enteral nutrition on a daily basis during the NICU stay, including the source of breast milk (mother's own milk and/or donor milk), total volume and the desired level of fortification based on individual analyses of the true protein content in the breast milk when such analyses have been done. The intervention is not blinded for the caregivers, clinical staff or study nurses. Targeted fortification will be applied at all study centres. The level of fortification is prescribed stepwise according to local guidelines in order to achieve appropriate protein intake and to ensure that intakes of all nutrients are within recommended ranges. The daily level of fortification for each infant is primarily based on protein intake and the aim is initially 4.0–4.5 g/kg/day but will gradually be decreased, when the infant is approaching term equivalent age and is showing adequate growth. The growth will be assessed by the physicians in charge on a daily basis guided by the same growth charts that are used by all participating NICUs and according to Swedish national guidelines. Macronutrient analyses of mother's own milk (MOM) are performed weekly using an infrared breast milk analyser (Miris, Uppsala, Sweden). Breast milk analyses of donor breast milk are performed once for each batch. If the supplementation with the breast milk fortifier starts before the first analysis has been done, half a dose of fortification is recommended during the first 1–3 days in both study groups. To assist in calculating the individual nutritional needs the computer-aided nutrition calculation programme Nutrium (Nutrium AB, Umeå, Sweden) is used. This will also be used in the prescription of other important supplements (eg, vitamins, iron, calcium and phosphorous) when needed. Since the human milk-based fortifier, that will be used in this trial, has a relatively high content of calcium and phosphorous (online supplemental table 1), the total daily volume of enteral feeds should not exceed 170 mL/kg/day when the breast milk has been fortified with a full dose. Fat supplements can be considered if energy intake is low and growth is suboptimal. The infants receiving the human milk-based nutrient fortifier are supplemented with the human milk-based caloric fortifier Humavant CR (online supplemental table 1), while the infants receiving standard bovine protein-based fortification are supplemented with the standard lipid products used at the unit. The infants in both groups must not be fed with formula during the intervention period, which ends at postmenstrual week 34+0. If protein fortification is still needed hereafter, there will be a transition period in the human milk-based group where the fortification of the breast milk is gradually substituted with standard bovine-based fortifier during a 5-day period.

## Discontinuation

If it is in the best interest of the subject, based on the responsible physician's discretion, a patient may be discontinued from the study. The legal guardians also have the right to discontinue participation in the study at

any time. The subject will still continue to be included in the study for clinical data collection, unless opposed by the legal guardians.

## Primary and secondary outcomes

The enrolled infants are characterised with clinical data including growth, feeding intolerance, use of enteral and parenteral nutrition, treatment, antibiotics and complications collected daily in a study specific case report form (CRF) from birth until discharge from the hospital (not longer than postmenstrual week 44+0).

Primary and secondary clinical outcomes are listed in table 1. The primary outcome is a composite of NEC stage II–III,[33] culture-proven sepsis and mortality during the study period. For the NEC diagnosis, radiological assessment will be made blinded by independent radiologists. Biopsies from the intestine will be collected if acute surgery is performed. If surgery confirms NEC, there is no need of positive radiological findings for diagnosis. The final decision is confirmed by a blinded consensus panel review consisting of the investigators. If the NEC diagnosis is confirmed, this diagnosis will replace any previous sepsis diagnosis during the duration of the NEC episode. For the diagnosis of culture-proven sepsis a positive blood, urine or cerebrospinal fluid culture is required. Furthermore, both clinical deterioration and a laboratory inflammatory response (white blood cell count <5 or >$20\times10^9$ cells/L or total platelet count <$100\times10^9$ cells/L or C-reactive protein >15 mg/L) are required to fulfil the criteria of culture-proven sepsis.[34] All data on nutrition and other fluids will be entered daily in the Nutrium software (Nutrium AB, Umeå, Sweden).

Associated predictor variables (covariates) are listed in table 2.

A follow-up focusing on neurological development, growth and feeding problems will be performed at 2 years (±3 months) of age (corrected) and 5.5 years (±3 months) of age (uncorrected). Additional data from the 2 and 5.5-year follow-up will be obtained from the Swedish neonatal quality register (www.snq.se).

Stool, urine, blood and breast milk samples are collected and stored in a longitudinal manner for microbiology, metabolomic, proteomic, lipidomic and immunology analyses in order to study underlying mechanisms. A separate protocol for each laboratory analysis will be created. Time points of collection of samples are displayed in figure 1. Furthermore, vitamin and micronutrient blood levels will be analysed in a sample of infants from both study groups. Blood, stool, urine and breast milk samples are included in the biobank at Children and Women centre at the University Hospital in Linköping, County of Östergötland, Sweden.

## Safety analyses and the data and safety monitoring board (DSMB)

Study monitoring is made by Fravil Clinical Consulting, Stockholm, Sweden, which is independent from the sponsor and competing interest. Moderate and severe adverse events (SAE) until discharge will be recorded. Many adverse events are also considered as clinical outcomes and displayed in table 1. Other adverse events that will be recorded are displayed in box 2. The morbidity in extremely preterm infants is very high. Thus, a high incidence of severe and moderate adverse events unrelated to the study product could be expected in the participating infants. Typical severe conditions affecting preterm infants are listed in box 2. The incidence of these SAE will be assessed by an independent DSMB to make interim safety analyses after 50, 100 and 150 completed CRFs until discharge. If any SAE is significantly more common (p<0.05) in the human milk-based as compared with the standard group, the DSMB will bring in all medical data on infants affected by this specific SAE and assess the causality for the specific SAE in the affected infants. Based on this analysis, the DSMB will decide if the trial can continue or not after consultation with the coordinating principal investigator and the sponsor. The DSMB may decide that it is ethically correct to pursue if the positive effects of the active intervention outweigh the SAE. In addition, the investigator or the attending physicians at the study site are required to report any suspected unexpected severe adverse reaction (SUSAR) to the sponsor within 24 hours. The sponsor will report SUSARs to the manufacturer and the DSMB.

## Data analysis and statistics

### Estimated sample size and power

The composite of NEC, culture-proven sepsis and mortality was 47% in extremely preterm infants surviving 3 days according to the Swedish neonatal quality register (www.snq.se). Since there is no well-powered trial with 100% coverage of breast milk, it is difficult to estimate the effect on NEC and sepsis from previous trials. With a conservative estimation, based on published studies,[29 30 35] the incidence of the composite outcome is estimated to be reduced to 28.0%. With at least 101 infants in each group a reduction from 47% in the control to 28% in the active group would be detected at a 5% level of significance and with 80% power. With a dropout rate of 10% a total of 222 subjects are needed.

Due to uncertainties in the presupposed effect size, conditional power is estimated in the following sense. An evaluation of the overall rate of the primary clinical endpoint of NEC/sepsis/death will be made prior to the formal analysis in order to determine whether the trial sample size should be re-evaluated and increased in order to continue study enrolment. This evaluation will be based on the methodology suggested by Gould.[36] An independent statistician, not associated with the study conduct will perform a sample size re-estimation when 150 infants have fulfilled the study period. The definitive sample size might be increased (never decreased) based on this interim analysis. Evidently, there is a need of a clinically relevant upper limit of the number of included infants. With a conservative estimation of a 50% NEC reduction in the human milk-based group, a reduction

**Table 1** Clinical outcomes

| | Description | Time frame |
|---|---|---|
| ***Primary clinical outcome*** | | |
| Composite of necrotising enterocolitis (NEC), culture-proven sepsis and mortality | Requires any of these three diagnoses to fulfil the criterion | A |
| ***Secondary clinical outcomes*** | | |
| Feeding intolerance and growth | | |
| Time to reach full enteral feeds | The day of life the infant has received at least 150 mL/kg enteral feeds | A |
| Feeding interruption | Number of days feedings held for ≥12 hours or feeds reduced by >50% (mL/kg/day) not due to a clinical procedure or transitioning to the breast | A |
| Parenteral nutrition | Number of days of parental amino acid and/or lipid infusion; only days when the enteral feed <150 mL/kg/day should be included | A |
| Gastric aspirates | ≥100% of prefeed volume (2 hours volume if continuous); lower limit of 2 mL/kg | A |
| Stool frequency | Number of stools per day | A |
| Time to regain birth weight | First day of three success days when the infant weight is greater than birth weight | A |
| Weight, height and head circumference | SD score will be used for the calculations | B |
| Clinical variables for morbidity | | |
| Infancy | | |
| Mortality | | A |
| NEC | Bell's stage II–III;[33] radiological assessment will be made blinded by an independent radiologist; the final decision will be confirmed by a blinded consensus panel review consisting of the investigators | A |
| Culture-proven sepsis | Positive blood- and/or urine- and/or CSF culture, clinical deterioration and laboratory inflammatory response;[34] further classified into early (<72 hours postpartum) or late (>72 hours postpartum) onset. | A |
| Composite of NEC and culture-proven sepsis | Requires any of these two diagnoses to fulfil the criterion | A |
| Mortality and morbidity index | Composite measure requiring any of the following: death, NEC stage II–III, culture-proven sepsis, moderate to severe BPD or ROP stage III–V | A |
| Spontaneous intestinal perforation | Intestinal perforation without signs of intramural and/or portal gas and no signs of inflammation at surgery | A |
| Abdominal surgery | | A |
| Suspected sepsis, not culture-proven | Clinical deterioration and laboratory inflammatory response but negative blood culture | A |
| Pneumonia | Pathological X-ray confirmed by a radiologist, need of increased respiratory support/oxygen and laboratory inflammatory response | A |
| Bronchopulmonary dysplasia (BPD) | Need of extra oxygen, high flow nasal cannula, CPAP or ventilator at w36+0[40] | C |
| Retinopathy of prematurity (ROP) | Diagnosed by an independent ophthalmologist according to international classification; classified into stage I–V;[41] the diagnosis is set after w42+0 | D |
| Intraventricular haemorrhage | Assessed with ultrasound; classified into grade I–IV[42] | A |

Continued

**Table 1** Continued

| | Description | Time frame |
|---|---|---|
| Periventricular leukomalacia | Assessed with ultrasound and MRI; criteria according to de Vries[43] | A |
| Number of days with intensive care | Need of respirator or CPAP | A |
| Length of stay at the hospital | GW and day at discharge | A |
| Length of need of feeding tube | GW and day when the infant does not need it anymore | A |
| 2-year follow-up | | |
| Neurocognitive development | Bayleys III,[44] Parent Report of Children's Abilities–Revised (PARCA-R), and Ages and Stages Questionnaires (ASQ-3) tested by psychologist | E |
| Cerebral palsy | Tested by an experienced paediatrician and/or physiotherapist | E |
| Epilepsy | | E |
| Strabismus and/or impaired vision | | E |
| Impaired hearing | | E |
| Respiratory support | Need of extra oxygen and/or ventilatory support | F |
| Wheeze and/or asthma | | G |
| Severe infections | | H |
| Mortality | Including cause of death | H |
| Need of feeding tube | | H |
| Extra nutritional support | | H |
| Level of education of the parents | | E |
| Family status | | E |
| Day-care | | E |
| 5.5-year follow-up | | |
| Neurocognitive development | Wechsler Preschool and Primary Scale of Intelligence IV tested by a psychologist and Movement ABC-2 tested by a physiotherapist | I |
| Cerebral palsy | Tested by an experienced paediatrician and/or physiotherapist | I |
| Epilepsy | | I |
| Strabismus and/or impaired vision | | I |
| Impaired hearing | | I |
| Wheeze and/or asthma | | I |
| Mortality | Including cause of death | J |

Timeframe A, from birth until discharge (no longer than w44+0); B, at 7, 14, 21 and 28 days, end of intervention (≤ w34+0), w36+0, discharge (no longer than w44+0), and 2 years of age (corrected) and 5.5 years of age (uncorrected); C, at w36+0; D, from birth until w42+0; E, at 2 years of age (corrected); F, from w44+0 until 2 years of age; G, from birth until 2 years of age; H, from discharge or w44+0 (whether comes first) until 2 years of age; I, at 5.5 years of age (uncorrected); J, from 2 years until 5 years of age. Week (w) refers to postmenstrual week. CPAP, continuous positive airway pressure; CSF, cerebrospinal fluid.

| Table 2 | Associated predictor variables (covariates) |
| --- | --- |
| 1 | Gender |
| 2 | Caesarean section |
| 3 | Multiple pregnancies |
| 4 | Birth weight and height |
| 5 | Small for gestational age, birth weight<2 SD |
| 6 | Maternal smoking during pregnancy |
| 7 | Preeclampsia, diagnosis by the responsible obstetrician |
| 8 | Chorioamnionitis, clinical diagnosis by the responsible obstetrician |
| 9 | Preterm premature rupture of membranes, rupture>1 hour before contractions started |
| 10 | Antenatal antibiotics, pertaining the period of the mother's actual attendance at the hospital |
| 11 | Antenatal corticosteroids; the mother should have received at least 12 mg betamethasone. The corticosteroid prophylaxis is considered completed if the mother has received two doses at least 24 hours before delivery |
| 12 | Born at level 1–2 NICUs |
| 13 | NICU inclusion site |
| 14 | Apgar score |
| 15 | Surfactant-administration |
| 16 | Intubation |
| 17 | Infant respiratory distress syndrome, verified by X-ray |
| 18 | Mechanical ventilation, duration |
| 19 | Patent ductus arteriosus, requiring medical or surgical treatment |
| 20 | Antibiotics, drug, treatment period and number of days |
| 21 | Probiotics, name, treatment period and number of days |
| 22 | Opioids, drug, treatment period and number of days |
| 23 | Gastric acid inhibitors, drug, treatment period and number of days |
| 24 | Day of life when the supplementation of the study product was started |
| 25 | The amount of enteral feeds per day that the infants received when the supplementation of the study product was started |
| 26 | Number of days the infant has not received the study product |
| 27 | Intravenous lines, number of days |
| 28 | Insulin treatment, number of days |
| 29 | Postnatal corticosteroids, number of days |
| 30 | Inotropic drugs, number of days |
| 31 | Feeding regime, continuous or bolus |
| 32 | Amount of nutrient protein fortifier per day |
| 33 | Amount of fat supplement per day |

Continued

| Table 2 | Continued |
| --- | --- |
| 34 | Total amount of protein, fat, carbohydrate, energy and micronutrient per day |
| 35 | The relative amount of donor breast milk per day |
| 36 | Amount of extra enteral lipid |
| 37 | Breastfeeding, exclusive or partial |

NICU, neonatal intensive care unit.

of the composite outcome from 47% in the control to 31% in the active group would be detected at a 5% level of significance and 80% power with 145 infants in each group. Based on this estimation, the upper limit of included infants in this study is 322 allowing for an approximate 10% dropout rate.

The secondary outcome composite measure of culture-proven sepsis and NEC could be of special interest, as it is not affected by mortality rates, which legitimated a specific estimation of the number to include. The incidence of this outcome was 42% (www.snq.se) and could be estimated to be reduced to 26% based on the background data described above. With at least 142 infants in each group, a reduction from 42% to 26% in the active group would be detected at a 5% level of significance, 80% power and a dropout rate of 10%. A total of 312 infants would be needed for this outcome. A similar evaluation will also be made for this outcome after 150 infants have completed the study in order to determine whether the trial sample size should be re-evaluated and increased (to a maximum of 322 infants).

The power of the lab analyses will not be estimated in advance, but with at least 30 cases of the clinical outcome (eg, NEC) we expect to have enough power to show relevant difference between groups.

The primary basis for all analyses of the clinical outcome will be the intention-to-treat paradigm. Only outcomes with an onset after the inclusion will be included in these analyses. Systematic bias due to drop-outs will be controlled with missing data analyses. Multiple imputation analyses will be performed if there will be a drop-out of >10%. Secondary per protocol analyses will also be made, including only infants receiving fortification and completing the intervention.

In case of a very strong effect of the active treatment, the study can be prematurely terminated based on decision by the sponsor and the DSMB, if the primary outcome is significantly lower (with a significance level <0.001) in the human milk-based group than in the standard fortification group in the interim analysis made after 150 infants have been included.[37 38] If the significance level is ≥0.001 the study enrolment will continue.

## Statistical methods
The primary outcome in the clinical trial and other categorical variables will be analysed with $\chi^2$ test and adjustments for potential confounders will be made with

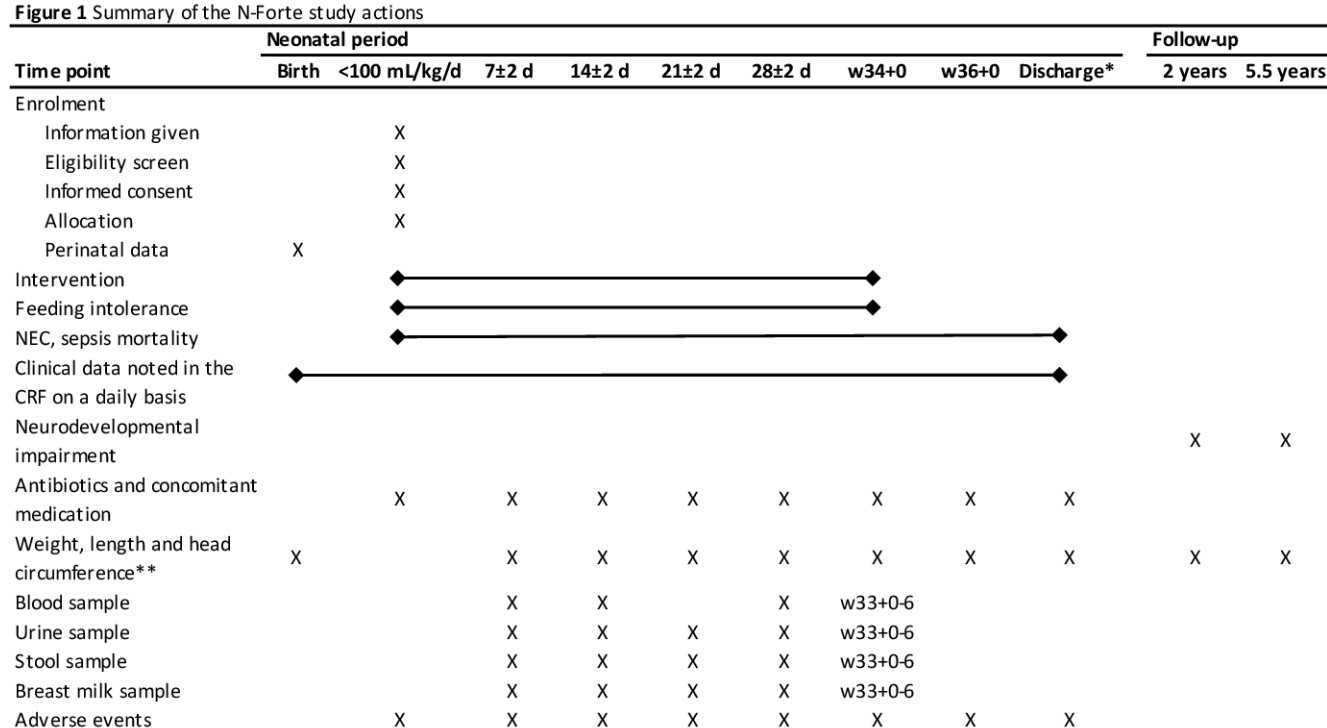

**Figure 1** Summary of the N-Forte study actions. CRF, case report form; NEC, necrotising enterocolitis.

logistic regression. Continuous variables will be analysed with Student's t-test if the distribution is normal and Mann-Whitney U test if not. Repeated measures-analysis of variance will be used for longitudinal data. Baseline characteristics will be summarised by means and SD for continuous data and counts and percentages for categorical data. The statistical discrimination will be at significance level of 0.05. Bioinformatic tools will be applied for high-throughput data. Distributions between groups will be statistically tested using the analysis of similarities. Principal component analyses will be performed to display beta-diversity, and UniFrac to analyse differences in beta-diversity between the groups. Alpha diversity will be calculated using Shannon's diversity index. False discovery rate correction will be made due to multiple comparisons (q<0.05).

### Patient and public involvement
The Swedish patient organisation Svenska Prematurförbundet was involved in the planning of the study design.

### Time frame for the study actions
Recruitment to the N-forte study began in 2019 and is planned to be completed during 2022. The study time points are presented in figure 1.

### DISCUSSION
There is still an important gap of evidence in this field, since there is no well-powered trial in extremely preterm infants comparing a human milk-based fortifier with bovine milk-based fortifier in a setting where both study groups are fed exclusively with breast milk. Such a trial is needed to provide evidence if a human milk-based fortifier is superior to a bovine-based one. Moreover, due to the introduction of active interventions for infants also born before gestational week 25 during the last decades,[1] a trial in Nordic centres will include a substantial number of infants born in gestational week 22–25, a patient population that could be expected to gain the most of a diet free from bovine protein.

This is a prospective randomised controlled trial to achieve the highest level of evidence. However, it is not blinded as it would not be possible to prescribe

---

**Box 2   Expected severe adverse events (SAE)**

► Severe infection such as pneumonia, sepsis or meningitis
► CMV infection, upper respiratory viral infection
► NEC, SIP and/or need of abdominal surgery
► Respiratory distress syndrome (RDS) and BPD
► Intracranial bleeding, PVL or hydrocephalus
► Lung bleeding
► Pneumothorax, pleural effusion
► Pulmonary hypertension
► Persistent ductus arteriosus (PDA)
► Retinopathy of prematurity (ROP)
► Death

BPD, bronchopulmonary dysplasia; CMV, cytomegalovirus; NEC, necrotising enterocolitis; PVL, periventricular leukomalacia; SIP, spontaneous intestinal perforation.

the fortifier and prepare the breast milk in a blinded fashion. First, the prescriptions are based on individually targeted fortifications on a daily basis by different physicians. It is essential that the clinician or the dietician knows which study group the individual patient belongs to, since the fortifiers are not exactly equal in nutrient content. Conventional fortifiers are powders mixed into the breast milk whereas human milk-based fortifiers are in liquid form and thereby substituting a part of the breastmilk given. Second, conducting centralised analyses and distribution of fortified breast milk[31] are not an option for logistical reasons as this is a multicentre study with more than 20 participating hospitals all over Sweden.

The criteria for the primary outcomes of culture-proven sepsis and NEC are therefore objective. The radiological assessment of NEC will be made blinded by an independent radiologist. The secondary outcome ROP will be diagnosed by an independent ophthalmologist. There are, however, outcomes, that potentially could be affected by the unblinded design of the trial, such as feeding intolerance. The results for these outcomes will therefore be interpreted with great caution.

Primary endpoint will be a composite of NEC, culture-proven sepsis and death. The rationale for the composite variable is that NEC and sepsis share many pathogenic mechanisms and that the diagnosis of NEC and sepsis often is a continuum. Furthermore, mortality constitutes an intrinsic censoring effect in infants at high risk of developing severe sepsis or NEC. In parallel, it is also logical to introduce a mortality and morbidity index (composite measure requiring any of the following: death, NEC stage II–III, culture-proven sepsis, moderate to severe BPD or ROP stage III–V) as a secondary outcome as shown in table 1. Only conditions occurring after the inclusion will be included for this outcome; hence, early debuting conditions like for example, intraventricular haemorrhage and early-onset sepsis (<72 hours) are not included.

Macronutrient analyses of MOM are performed weekly for the targeted fortification. It has been argued that breast milk analyses should be done on a daily basis because of the day-to-day variations.[39] Such approach is not clinically feasible as it will consume far too much of MOM milk to be ethical justified. Consequently, the protein and energy intake by the neonates fed exclusively or mainly own mother's milk may be out of the target levels, while this will not apply for infants mainly fed by donor milk, since all donor milk will be analysed. These differences are not expected to have a significant impact on the primary outcome measures, especially not when the study groups are randomised, but they may affect some of the secondary outcomes, such as growth and metabolomics. However, we will have thorough data on the amount of donor milk the infant has received each day, which then will be taken in account in subsequent analyses.

## ETHICS AND DISSEMINATION

Research addressing ELGA infants is crucial in order to achieve knowledge of possible causes underlying the development of severe complications and finding preventive strategies. Intervention studies are needed to refute or confirm the suggested effect of human milk-based nutrient fortification. We do not perceive any major health hazard with the study design. Human milk-based nutrient fortifiers are considered to be safe in ELGA infants and safety will be closely monitored.

A written informed consent will be obtained from legal guardians before inclusion. Infants or their families have no specific benefit of study participation besides the possible effects of the human milk-based nutrient fortifier in the active group. Besides the supplementation of the breast milk fortifier and the samples collected in the trial, the participating infants will not be treated differently compared with routine care. The potential benefit for future ELGA infants, however, could be substantial. Finding an intervention that reduces NEC, sepsis and mortality would have a major impact of the well-being in this patient population. It may also elicit beneficial effects on health economics. Only randomised clinical trials will give sufficient evidence for a general recommendation of a new treatment.

Personal information about enrolled participants will be collected, shared and maintained in accordance with the EU General Data Protection Regulation.

Crown Princess Victoria Children's Hospital, County of Östergötland, Linköping, Sweden, is the sponsor and owns all the information obtained in the trial together with the co-ordinating principal investigator. The study is made in collaboration with the company Prolacta Bioscience producing the human milk-based fortifier, which is, for this study, contributed by Prolacta. The study, however, is investigator-initiated. There are no publication restrictions. Study results will be presented at relevant conferences and submitted to peer-reviewed journals. None of the investigators have a financial interest in Prolacta Bioscience. The donating mother signs an informed consent form stating the rights and responsibilities of the donating mother as well as the payment for the provided breast milk. Only excess breast milk, produced by the donating mother, that is beyond the consumptive needs of her nursing child, is accepted.

The potential benefits are considered to outweigh the possible discomfort to the infants and their families. The study is conducted according to ICH/GCP guidelines and was approved by the regional ethical review board at Linköping University (Dnr 2018/193-31, Dnr 2018/384-32).

**Author affiliations**
[1]Department of Biomedical and Clinical Sciences, Linköping University, Linköping, Sweden
[2]Department of Paediatrics, Linköping University, Linköping, Östergötland, Sweden
[3]Department of Women's and Children's Health, Uppsala University, Uppsala, Sweden
[4]Department of Clinical Sciences, Pediatrics, Umeå University, Umeå, Sweden

⁵Institute of Clinical Sciences, Department of Paediatrics, Sahlgrenska Academy, Göteborg, Sweden
⁶Queen Silvia Children's Hospital, Department of Paediatrics, Sahlgrenska University Hospital, Göteborg, Sweden
⁷Department of Clinical Science, Intervention, and Technology, Karolinska Institutet, Stockholm, Sweden
⁸Department of Neonatology, Karolinska University Hospital, Stockholm, Sweden

**Acknowledgements** The authors wish to thank all participating hospitals and staff members and the Swedish patient organisation Svenska Prematurförbundet for their invaluable contributions in the planning and realisation of the study.

**Contributors** Coordinating principal investigator: TA. Conceptualisation: TA, FA, MD, AE. Formal analysis: TA, MD. Funding acquisition: TA. Methodology: TA, FA, MD, AE. Project administration (steering committee): TA, FA, GBJ, MD, AE, LN. Resources: TA, FA, MD, AE, LN. Supervision: TA. Validation: TA, FA, MD, AE, LN. Writing – original draft: GBJ. Writing – review and editing: TA, FA, GBJ, MD, AE, LN.

**Funding** This work is funded by grants from the Swedish Research Council (2020-01111 and 2019-01005), the Research Council for Southeast Sweden, ALF Grants, Prolacta Bioscience, CA, USA.

**Competing interests** TA has got a grant for the present study by Prolacta Bioscience, CA, USA, but none of the investigators have any financial interest in Prolacta Bioscience.

**Patient consent for publication** Not applicable.

**Provenance and peer review** Not commissioned; externally peer reviewed.

**ORCID iDs**
Georg Bach Jensen http://orcid.org/0000-0002-2905-7698
Fredrik Ahlsson http://orcid.org/0000-0002-8413-9274
Magnus Domellöf http://orcid.org/0000-0002-0726-7029
Anders Elfvin http://orcid.org/0000-0002-1912-9563
Lars Naver http://orcid.org/0000-0001-6027-0211
Thomas Abrahamsson http://orcid.org/0000-0002-0190-8294

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
