## [Reviewer comments · BMJ Open]

ARTICLE DETAILS

TITLE (PROVISIONAL)	Nordic study on human milk fortification in extremely preterm infants (N forte): the study protocol of a randomised controlled trial
AUTHORS	Jensen, Georg Bach; Ahlsson, Fredrik; Domellöf, Magnus; Elfvin, Anders; Naver, Lars; Abrahamsson, Thomas

VERSION 1 – REVIEW

REVIEWER	Perrin, Maryanne University of North Carolina at Greensboro
REVIEW RETURNED	28-Jun-2021

GENERAL COMMENTS	Summary: This is the protocol for a multi-center trial to determine the impact of human-based versus bovine-based fortifiers in human-milk fed (MOM or DHM) preterm infants born before 28 weeks gestational age. The study treatment/control are delivered through postmenstrual week 34+0 and the main outcome measures are NEC, sepsis, and death. Biosamples of stool, urine, blood and breast milk will be collected to explore potential mechanisms. Neurodevelopment and growth will be assessed at 2 years and 5.5 years. There are limited longitudinal studies on the impact of fortifier type in human milk-fed preterm infants, and additional studies are needed to follow up on the findings from the OptiMoM trial conducted in Canada which found no difference in feeding intolerance. The OptiMoM study was not powered to detect differences in NEC. Important strengths of the N-forte study compared to prior research include: (1) an emphasis on infants 23-25 weeks gestational age who may benefit most from this intervention; (2) the use of targeted fortification methods based on macronutrient analysis of breast milk so that nutrient goals are more likely achieved (however, a consequence of the milk analysis is that clinicians will not be blinded to feeding type); (3) the collection of multiple biospecimens for future mechanistic exploration. Page 4, line 10. Since reference #29 and #30 also used bovine-based formulas in addition to bovine-based fortifiers, these studies were not actually designed to examine the impact of fortifiers, but would be more accurately described as “examining the impact of bovine-based proteins”. Only O’Connor (ref #31) was designed to evaluate the impact of the fortifier. Page 5, line 41. Are all the bovine-based fortifiers liquid, or do some NICUs use a powdered bovine fortifier? Page 5, line 48-49. Please clarify if target fortification will be based on total or true protein values, since the Miris reports both.
--

	Page 6, line 11. Do all facilities use the same definition of sub-optimal growth where additional fortification of milk is initiated? Please define. Page 6, line 16. This sentence is difficult to understand with the double negative. Do you mean that the infants in both groups must not be fed with bovine formula? Page 6, line 60. In your published papers it will be important to describe the collection protocol for the breast milk samples as the collection method can have a profound impact on the fat/calories in the sample. Page 11. Please clarify if the human milk fortifier was purchased from Prolacta or if they contributed it for the study. Table 3. The list of proposed covariates is extensive. Should you also include NICU location as a covariate? What about social determinants of health including race and income?
--	--

REVIEWER	Agakidou, Eleni Ippokration General Hospital, 1st Dept of Neonatology and NICU
REVIEW RETURNED	02-Aug-2021

GENERAL COMMENTS	This is a multicenter, randomized controlled study investigating the potential superiority of the use of a human milk – derived fortifier over the conventional bovine fortifier for feeding extremely low gestational age (ELGA) neonates with breast milk. This is an interesting study which may clarify whether fortification of breast milk (own mother’s milk or bank milk) using a human milk – derived fortifier is associated with a significant decrease in the incidence of necrotizing enterocolitis, sepsis, and mortality as well as with the growth and occurrence of other short- and long-term morbidities, compared to the use of bovine milk-derived fortifier. An important strength of the study is the inclusion of ELGA neonates, a population being at a highest risk of NEC and sepsis and high mortality, for which there are only limited relevant published data. In addition, the study is well designed and presented while complying to the SPIRIT recommendations for interventional trials. There are only two issues regarding methodology that need attention:  1. The weekly assessment of own mother’s milk protein and energy content may not be enough to ensure the desired macronutrient intake, considering the day-to-day variability of maternal milk composition (Rochow N, Fusch G, Zapanta B, Ali A, Barui S, Fusch C. Target fortification of breast milk: how often should milk analysis be done? Nutrients. 2015 Apr 1;7(4):2297-310. doi: 10.3390/nu7042297.). Consequently, the protein and energy intake by the neonates fed exclusively or mainly own mother’s milk may be out of the target levels. As this problem does not apply to the donor milk – fed neonates, it may result to unpredictable differences in protein and energy intake between the two study groups. Although, these differences are not expected to have a significant impact on the primary outcome measures, they may affect some of the secondary outcomes, such as growth and metabolomics/proteomics. 2. Table 3. The predictive variables (n=36), which obviously will be included in the multiple regression models, are too many for the
---

	number of infants that will be studied (a total of 322 allowing a 10% dropout rate). It has been suggested that at least 10 subjects are needed for every variable included in the regression model. The authors could overcome this obstacle by combining certain variables.
--	--

VERSION 1 – AUTHOR RESPONSE

Reviewer: 1

Comments to the Author: “Summary: This is the protocol for a multi-center trial to determine the impact of human-based versus bovine-based fortifiers in human-milk fed (MOM or DHM) preterm infants born before 28 weeks gestational age. The study treatment/control are delivered through postmenstrual week 34+0 and the main outcome measures are NEC, sepsis, and death. Biosamples of stool, urine, blood and breast milk will be collected to explore potential mechanisms. Neurodevelopment and growth will be assessed at 2 years and 5.5 years. There are limited longitudinal studies on the impact of fortifier type in human milk-fed preterm infants, and additional studies are needed to follow up on the findings from the OptiMoM trial conducted in Canada which found no difference in feeding intolerance. The OptiMoM study was not powered to detect differences in NEC. Important strengths of the N-forte study compared to prior research include: (1) an emphasis on infants 23-25 weeks gestational age who may benefit most from this intervention; (2) the use of targeted fortification methods based on macronutrient analysis of breast milk so that nutrient goals are more likely achieved (however, a consequence of the milk analysis is that clinicians will not be blinded to feeding type); (3) the collection of multiple biospecimens for future mechanistic exploration”

- Reply: Thanks for the appreciative comments.

Comment: “Page 4, line 10. Since reference #29 and #30 also used bovine-based formulas in addition to bovine-based fortifiers, these studies were not actually designed to examine the impact of fortifiers, but would be more accurately described as “examining the impact of bovine-based proteins”. Only O’Connor (ref #31) was designed to evaluate the impact of the fortifier.”

- Reply: We agree and have made changes in the manuscript accordingly.

Comment: “Page 5, line 41. Are all the bovine-based fortifiers liquid, or do some NICUs use a powdered bovine fortifier?”

- Reply: The bovine-based fortifiers are all powdered. We mention this in the discussion section when we discuss the differences between the human- and the bovine-based fortifiers in the trial.

Comment: “Page 5, line 48-49. Please clarify if target fortification will be based on total or true protein values, since the Miris reports both.”

- Reply: True protein values are used which has been added to the manuscript as suggested.

Comment: “Page 6, line 11. Do all facilities use the same definition of sub-optimal growth where additional fortification of milk is initiated? Please define.”

- Reply: The growth will be assessed by the physicians and dietitians and is based on the growth charts used by all facilities considering relevant clinical parameters on an individual level and according to Swedish national guidelines. A comment has been made in the manuscript (p. 5) regarding this.

Comment: “Page 6, line 16. This sentence is difficult to understand with the double negative. Do you mean that the infants in both groups must not be fed with bovine formula?”

- Reply: Yes, no infant, whatever group, should be fed with formula. Only breast milk, donated or mother’s own. The sentence has been rephrased.

Comment: “Page 6, line 60. In your published papers it will be important to describe the collection protocol for the breast milk samples as the collection method can have a profound impact on the fat/calories in the sample.”

- Reply: We agree. Breast milk samples are only collected if the mother has an excess of it. The sample is collected from the breast milk that the infant will be fed with at the actual day (after fortification): at 1, 2, 3 and 4 weeks (± 2 days) of age and at postmenstrual week 33+0 – 33+6 in sterile 10 mL tubes. The sample is stored in -20°C . As soon as possible they are then transferred frozen to a -70°C freezer.

Comment: “Page 11. Please clarify if the human milk fortifier was purchased from Prolacta or if they contributed it for the study.”

- Reply: The human milk-based fortifier is contributed by Prolacta. This has now been stated in the Ethics section.

Comment: “Table 3. The list of proposed covariates is extensive. Should you also include NICU location as a covariate? What about social determinants of health including race and income?”

- Reply: “NICU inclusion site” has now been added to the list. We agree that social determinants could have been interesting, although the effects could be expected to be limited in Sweden, where all inhabitants will get the same standard of care, especially in the neonatal ward. Regarding income, this is not data that is asked for and included in the medical record in Sweden. By GDPR reasons, we decided not to ask specifically and include this potential sensitive data in our datafile. We concluded that the risks outweighed the benefits. Regarding race, it is, for ethical reasons, not possible to gather this information in Sweden.

Reviewer: 2

Comments to the Author: “This is a multicenter, randomized controlled study investigating the potential superiority of the use of a human milk – derived fortifier over the conventional bovine fortifier for feeding extremely low gestational age (ELGA) neonates with breast milk. This is an interesting study which may clarify whether fortification of breast milk (own mother’s milk or bank milk) using a human milk – derived fortifier is associated with a significant decrease in the incidence of necrotizing enterocolitis, sepsis, and mortality as well as with the growth and occurrence of other short- and long-term morbidities, compared to the use of bovine milk-derived fortifier. An important strength of the study is the inclusion of ELGA neonates, a population being at a highest risk of NEC and sepsis and high mortality, for which there are only limited relevant published data. In addition, the study is well designed and presented while complying to the SPIRIT recommendations for interventional trials.”

- Reply: We thank the reviewer for the appreciative comments and for recognising the importance of this study in high-risk ELGA neonates.

Reviewer 2 brings to attention two issues regarding methodology:

Comment: “1. The weekly assessment of own mother’s milk protein and energy content may not be enough to ensure the desired macronutrient intake, considering the day-to-day variability of maternal milk composition (Rochow N, Fusch G, Zapanta B, Ali A, Barui S, Fusch C. Target fortification of breast milk: how often should milk analysis be done? *Nutrients*. 2015 Apr 1;7(4):2297-310. doi: 10.3390/nu7042297.). Consequently, the protein and energy intake by the neonates fed exclusively or mainly own mother’s milk may be out of the target levels. As this problem does not apply to the donor milk – fed neonates, it may result to unpredictable differences in protein and energy intake between the two study groups. Although, these differences are not expected to have a significant impact on the primary outcome measures, they may affect some of the secondary outcomes, such as growth and metabolomics/proteomics.”

- Reply: We appreciate the comment. We agree that it may be beneficial to analyse the breast milk every day, but it would not have been feasible to change a well-established routine in 20 centres in Sweden for our trial. Moreover, we do not see those big differences between the different weeks in the same mother where the energy content over time is rather constant. There is a slow decrease in protein content during the first 4-6 weeks but no expected rapid day-to-day changes. As stated by the reviewers, this will probably not affect the primary outcome, especially not when the study is randomised. However, we have now added a comment on this issue in the Discussion section.
- Comment: “2. Table 3. The predictive variables (n=36), which obviously will be included in the multiple regression models, are too many for the number of infants that will be studied (a total of 322 allowing a 10% dropout rate). It has been suggested that at least 10 subjects are needed for every variable included in the regression model. The authors could overcome this obstacle by combining certain variables.”
- Reply: The predictive variables will be checked individually and add important information to the data. However, only potential confounders will be included in multivariate analyses. In general, only variables that significantly differ between groups will be considered as such potential confounders. To further limit the variables included to only be the relevant ones, directed acyclic graphs (DAGs) and other similar measures will also be used. Most of the covariates listed in Table 3 are expected to show non-significant effects and are thereby not expected to be included as potential confounders in the multivariate analysis of the primary outcome.